# What matters most to patients with severe aortic stenosis when choosing treatment? Framing the conversation for shared decision making

**Nananda F. Col[1]\*, Diana Otero[2]ᵒ, Brian R. Lindman[3]ᵒ, Aaron Horne[4], Melissa M. Levack[5], Long Ngo[6], Kimberly Goodloe[7], Susan Strong[8], Elvin Kaplan[9], Melissa Beaudry[10], Megan Coylewright[11]ᵒ**

1 Shared Decision Making Resources, Georgetown, ME and University of New England, Biddeford, Maine, United States of America, 2 Department of Cardiovascular Medicine, University of Louisville School of Medicine, Louisville, Kentucky, United States of America, 3 Structural Heart and Valve Center, Vanderbilt University Medical Center, Nashville, Tennessee, United States of America, 4 HeartCare Specialists, Medical City North Hills, North Richland Hills, Texas, United States of America, 5 Department of Cardiac Surgery, Vanderbilt University Medical Center, Nashville, Tennessee, United States of America, 6 Harvard Medical School, Boston, Massachusetts, United States of America, 7 American Heart Association Ambassador, Atlanta, Georgia, United States of America, 8 Heart Valve Voice US, Washington DC, United States of America, 9 Patient Collaborator, Brownsville, Vermont, United States of America, 10 Central Vermont Medical Center, Berlin, Vermont, United States of America, 11 Department of Cardiovascular Medicine, The Erlanger Heart and Lung Institute, Chattanooga, Tennessee, United States of America

ᵒ These authors contributed equally to this work.
\* Nananda@sdmresources.info

**Data Availability Statement:** De-identified data are in the following Dryad repository: https://doi.org/10.5061/dryad.2bvq83bsv or https://datadryad.

## Abstract

### Background

Guidelines recommend including the patient's values and preferences when choosing treatment for severe aortic stenosis (sAS). However, little is known about what matters most to patients as they develop treatment preferences. Our objective was to identify, prioritize, and organize patient-reported goals and features of treatment for sAS.

### Methods

This multi-center mixed-methods study conducted structured focus groups using the nominal group technique to identify patients' most important treatment goals and features. Patients separately rated and grouped those items using card sorting techniques. Multidimensional scaling and hierarchical cluster analyses generated a cognitive map and clusters.

### Results

51 adults with sAS and 3 caregivers with experience choosing treatment (age 36–92 years) were included. Participants were referred from multiple health centers across the U.S. and online. Eight nominal group meetings generated 32 unique treatment goals and 46 treatment features, which were grouped into 10 clusters of goals and 11 clusters of features. The

org/stash/share/yNP227NZm-6v9Ef8ARHY_
hzGdIlDOr6uAZG4rpUJqBs.

**Funding:** Financial support for this study was
provided by an independent grant from Edwards
Lifesciences. The funding agreement was through
Shared Decision Making Resources (NC) and
ensured the authors' independence in designing
the study, interpreting the data, writing, and
publishing the report.

**Competing interests:** NFC has received consulting
fees and research contracts from various entities
through her contract research organization Shared
Decision Making Resources. She received
independent research grants from Edwards
Lifesciences, Biogen, Pfizer, and EMD-Serono, and
has consulted for Janssen Pharmaceuticals. BRL
has served on the scientific advisory board for
Roche Diagnostics, has received research grants
from Edwards Lifesciences and Roche Diagnostics,
and has consulted for Medtronic. MC has received
consulting fees and research funding from Boston
Scientific and Edwards LifeSciences. MML has
received consulting fees from Boston Scientific,
Edwards LifeSciences, and Medtronics. This does
not alter our adherence to PLOS ONE policies on
sharing data and materials. There are no patents,
products in development or marketed products
associated with this research to declare.

most important clusters were: 1) trust in the healthcare team, 2) having good information
about options, and 3) long-term outlook. Other clusters addressed the need for and urgency
of treatment, being independent and active, overall health, quality of life, family and friends,
recovery, homecare, and the process of decision-making.

## Conclusions

These patient-reported items addressed the impact of the treatment decision on the lives of
patients and their families from the time of decision-making through recovery, homecare,
and beyond. Many attributes had not been previously reported for sAS. The goals and fea-
tures that patients' value, and the relative importance that they attach to them, differ from
those reported in clinical trials and vary substantially from one individual to another. These
findings are being used to design a shared decision-making tool to help patients and their cli-
nicians choose a treatment that aligns with the patients' priorities.

## Trial registration

ClinicalTrials.gov, Trial ID: NCT04755426, Trial URL https://clinicaltrials.gov/ct2/show/
NCT04755426.

## Introduction

Severe aortic stenosis (sAS) is a common heart valve disease that has increased in parallel with
the aging population worldwide [1]. Decisions about treatment are more complex with the
expanded availability of transcatheter aortic valve replacement (TAVR) in addition to tradi-
tional surgical aortic valve replacement (SAVR) across the surgical risk spectrum. While clini-
cal guidelines for valvular heart disease recommend shared decision making (SDM) that
incorporates patients' goals and values about treatment [2], SDM remains poorly imple-
mented, with values elicitation often neglected [3, 4].

Little is known about how patients with sAS compare their treatment options, which treat-
ment goals and features are most relevant, or factors that influence their treatment preferences
[5, 6]. There are few studies on patient preferences in structural heart disease and those studies
relied on preference elicitation methods that use predefined attributes or constructs that were
selected by expert physicians or investigators with little or no patient input [7–11]. Likewise,
benefit-risk analyses [12] and patient decision aids for sAS [4, 11, 13] focused on investigator-
identified outcomes defined in clinical trials such as mortality, need for permanent pacemaker,
vascular complications, and length of stay [2, 14, 15]. Those studies employed methods that
are easily influenced by the investigators' perspectives [9], with most focusing on a single pro-
cedure [7]. There are significant differences between healthcare providers' (HCP) and patients'
values and preferences [16, 17]. The validity of using investigator-identified outcomes to
understand how patients make decisions has not been established.

There is no consensus on how best to elicit patient treatment goals, preferences and values;
however, how they are elicited can influence the findings. To extract what matters most to
patients when choosing from a range of treatment options, the ideal method is one that elicits
the widest range of meaningful responses while minimizing investigator bias. Focus group dis-
cussions are often the default approach for gathering patient perspectives but are easily influ-
enced by the investigators' own perspectives. Furthermore, interactive group discussions can
influence participants' perspectives and inhibit creative thinking by pursuing a single train of

thought ('group think'); discourage participation of passive participants and those with differing perspectives; and promote premature evaluation that decreases the quality of the ideas generated in terms of creativity, originality, and practicality [18]. In contrast, the highly structured nominal group technique (NGT) has several benefits, even beyond minimizing investigator bias. The NGT methodology avoids "group think", prompts individuals to work independently to generate their own ideas, allows equal input by all participants, captures the language of participants, and objectively prioritizes and organizes findings [19, 20]. Participants of NGTs remain silent while responding to a focused, pretested question about their goals or values, with discussion limited to clarifying the question itself. Only after each person has independently recorded their responses do participants share their ideas with the group, with each person participating sequentially and equally. New responses stimulated by others' ideas are encouraged. Responses can be submitted anonymously to avoid self-censuring of embarrassing topics. A list of all responses from the group is generated and ranking of the most important items on the list is done silently and anonymously. Unlike focus group or structured interviews, NGT is a consensus method that generates responses to a question which can then be prioritized. Responses are not filtered through the lens of the investigator but are the verbatim responses of participants themselves. NGT participants are carefully selected to include the heterogeneity of the target audience [19, 20]. Card-sorting exercises captures how patients themselves organize the concepts generated by the NGTs, further diminishing investigator bias and elevating the patient voice.

Understanding what matters to patients with sAS is essential because patient goals and values drive their preferences for treatment. Furthermore, assumptions about what matters to patients influence the treatments clinicians offer to patients, how treatments are presented, and judgements about medical futility. These assumptions can contribute to underutilization of AVR [21] and socioeconomic, racial, and ethnic disparities in care [22]. Our objective was to use patient-centered methods to: (1) identify important patient-reported treatment goals and features for sAS, and (2) explore how patients prioritize and organize these goals and features when forming their preference for treatment. Such information will provide the building blocks for a SDM process that aims to meaningfully integrate the patient voice into treatment decisions.

## Methods

### Design

This multi-center mixed-methods study used an exploratory sequential design consisting of 8 nominal group technique (NGT) meetings coupled with 2 open card sorts ("cognitive mapping"), conducted between January and December 2020. The study was guided by a patient advisory group and approved by the Western Institutional Review Board (WIRB) Copernicus Group (WCG) Independent Review Board.

### Participants

We purposively recruited English-speaking adults with experience making decisions about sAS through their HCP or a study adviser with access to aortic stenosis advocacy organizations (American Heart Association Ambassadors, Heart Valve Voice US). We targeted patients who had experienced a treatment decision and its consequences in order to capture informed decisions that reflect the full range of considerations. We targeted diversity in socio-demographics and clinical factors, including those who choose medical therapy. Referring HCPs included interventional cardiologists, a cardiac surgeon, general cardiologists, and a nurse practitioner. Referral sites included medical centers in rural New Hampshire (Dartmouth-Hitchcock

Medical Center, Hanover, NH), Nashville, Tennessee (Vanderbilt University Medical Center), Louisville, KY (University of Louisville School of Medicine), Washington, DC (MedStar Washington Hospital Center), and rural Vermont (Central Vermont Medical Center, Berlin, VT). Participants provided informed consent.

## Study procedures

To elicit patients' goals and treatment features, NGT participants silently responded to one open-ended question about their goals or values (Fig 1). Participants in each group then shared each of their responses, clarified and consolidated responses into a list of unique items, and then privately ranked their top 9 items. Items were consolidated across NGT groups. To understand how participants organize these items, we conducted card sorting where participants (1) rated the importance of each goal or feature, using a 5-point Likert scale, (2) placed items into groups according to their perceived similarity, and (3) labeled their groups. In-person activities shifted online during the COVID-19 pandemic. An online asynchronous NGT protocol that mirrored the in-person protocol had been previously developed and validated [16, 23]. Surveys used customized Qualtrics© software.

The data underlying our findings are available through the Dryad depository at https://data dryad.org/stash/share/yNP227NZm-6v9Ef8ARHY_hzGdIlDOr6uAZG4rpUJqBs.

## Statistical analyses

We calculated the means (SD) and proportions of the importance ratings for each item and each cluster. Card sort data were transformed into a matrix according to how often two items were sorted into the same group. Multidimensional scaling (MDS) mapped the spatial relationships between these items [16]. We evaluated goodness-of-fit using the stress statistic, which indicates the differences between the observed and modeled data. Values < 0.15 indicate a good fit. Hierarchical cluster analysis (HCA) yielded a dendrogram tree whose branches depict possible clusters (S1 and S2 Figs in S1 Appendix). Patients then reviewed and labeled these clusters. Analyses used IBM SPSS Version 26. Heterogeneity analyses assessed the similarity among individual responses by comparing all possible pairs of responses. We compared

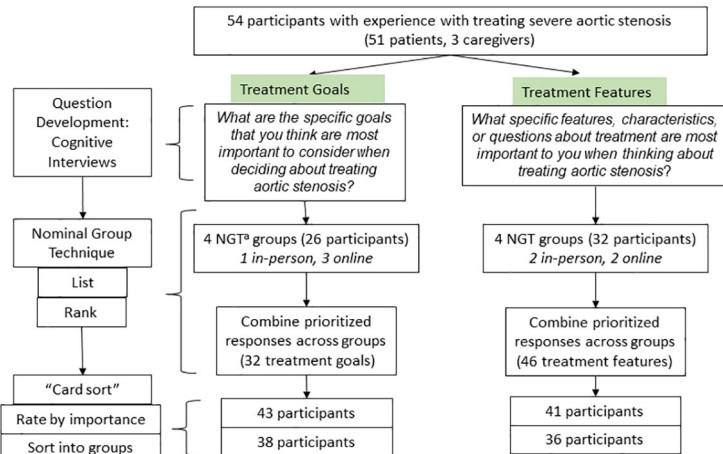

**Fig 1. Study design and sample.** This figure depicts the sequence of activities conducted during various stages of the study. Participants could participate in more than one study activity. Participants were enrolled through the course of the study. [a]NGT: Nominal Group Technique.

the ratings of each pair (32 goals, 46 features), calculating the mean of the absolute value of the paired difference to estimate the proportion of items that differed among pairs.

## Results

Study participants included 51 patients with sAS and 3 caregivers who participated in one or more study activities (Fig 1). Participants were 36 to 92 years old and 84.3% self-identified as white, while 43% were women Twelve percent completed high school or less, 17.6% completed some college, and 60.7% completed college; 19.6% reported low health literacy [24]. All participants had personally undergone decision-making regarding treatment of their sAS, or with their family member. Nearly two-thirds had a history of TAVR (62.7%), 27.5% SAVR, and 9.8% medical treatment. Patients had undergone AVR 0–11 years prior to participating. Patient characteristics are shown in Table 1. Most patients (92%) were referred by their HCP.

All the patient-reported goals and features elicited were highly rated by a substantial proportion of participants. The most highly endorsed goal (*To have a good medical team and facility*) was highly rated ("very important") by 90.7% and the least endorsed goal (*To choose an ethical treatment*) was highly rated by over 16%. There was substantial variability among individuals' ratings. Over 85% of participants' ratings differed from others by 50% or more.

**Table 1. Patient characteristics.**

| Characteristic | No. (%) (n = 51[a]) |
|---|---|
| Mean age, years (median, range) | 72.04 (72; 36–92) |
| Gender, n (%) | |
| Male | 28 (54.9%) |
| Female | 23 (43.1%) |
| **Race** n (%) | |
| White/Caucasian | 43 (84.3%) |
| Non-white, Black/African American, Multicultural | 8 (15.7%) |
| **Treatment** for Aortic Stenosis: | |
| TAVR[b] | 32 (62.7%) |
| SAVR[c] | 14 (27.5%) |
| Medical Treatment | 5 (9.8%) |
| **Year of diagnosis** in years, median (min, max) | 2012 (1984, 2020) |
| **Year of AVR**[d], median (min, max) | 2018 (2009, 2020) |
| **Education**, n (%) | |
| Less than high school | 3 (5.9%) |
| High school or GED | 3 (5.9%) |
| Some college | 9 (17.6%) |
| 2-year college or technical school | 4 (7.8%) |
| College graduate | 9 (17.6%) |
| Graduate school or professional degree | 22 (43.1%) |
| **Health literacy** (% with any challenge) | 14 (27.5%) |
| % with low reading health materials [24] | (19.6%) |

[a]Three caregivers of participating patients were included (all female, white).
[b]TAVR = Transcatheter Aortic Valve Replacement.
[c]SAVR = Surgical Aortic Valve Replacement.
[d]AVR = Aortic Valve Replacement.

## Patient-reported treatment goals

Four NGT meetings involving 26 participants generated 32 unique treatment goals. Card sorting organized these goals into 10 clusters and labeled each cluster (Table 2).

Clusters were named by the patient participants themselves to summarize the key attributes of the cluster. The patients' framing is listed in italics. The most important cluster was *To have trust in the doctor*, *medical team and hospital*, followed by *To have good information about my options* and *To lead a long life*. The most important goal, *To have a good medical team and facility*, was highly rated by 91% of respondents. *To lead a long life* was highly rated by *72.1%* and ranked ninth. Other important goals included *To be independent* and *To give my family peace of mind.*

The cognitive map illustrates how respondents organized the goals (Fig 2). The closer the items, the more often they were grouped together. Each point in the graph represents a linear combination of all 32 goals, mapped onto two dimensions, which we interpreted as a) timeline from decision-making to recovery and b) internal versus external factors. The stress statistic (0.077) indicates a good fit between the actual and modeled data.

## Patient-reported treatment features

Four NGT meetings involving 32 participants generated 46 unique treatment features. Card sorting identified 11 clusters (Table 3).

The most important cluster regarding treatment features, as opposed to goals, was *Qualities of the doctor and hospital*, followed by *Comparing my options*, *Long-term outlook*, and *If and when to have the procedure*. *Details about procedures and their complications* (which includes other heart problems such as atrial fibrillation, stroke or need for a pacemaker) was ranked 7th in importance. The most important individual treatment feature was *What are the options when replacing or repairing my valve*?, highly rated by 93%. Other important features included *What will recovery be like*, *How do I ease my fear or anxiety about choosing treatment* and *Will I feel pain*? The cognitive map is shown in Fig 3. Clusters relating to health-related quality of life overlap in the cognitive map (cluster 8, recovery and post-surgical care; cluster 5, impact on daily living and quality of life, and cluster 6, impact on overall health and medications). While statistical analyses identified either 1 large cluster (depicted in grey in Fig 3) or 3 sub-clusters (also shown in S4 Fig in S1 Appendix), patients with sAS partitioned this larger cluster into 3 distinct clusters rather than combine them into a larger category.

The dimensions of the treatment features map (Fig 3) were the same as the treatment goals map (a) timeline from decision-making to recovery and b) internal versus external factors). The stress statistic (0.094) indicates a good fit.

## Discussion

Using patient-centered methods, we identified a comprehensive set of patient-reported treatment goals and features, with minimal investigator input. These items addressed the impact of treatment for sAS on patients' lives from the time of decision-making through recovery and beyond. Patient-reported items tended to be vague about procedural complications (e.g., *fewer long-term risks*, *long-lasting solution*) but specific about feelings regarding their condition (e.g., fear, anxiety) and the aftermath of treatment (e.g. independence, recovery). Risks were often coupled to the feelings they evoked (e.g., *fear* of heart failure), and how procedures would be experienced (e.g., pain, being awake). Overall, we found that the goals and features that patients value differ from those reported in clinical trials and vary substantially from one individual to another.

**Table 2. Patient treatment goals, prioritized clusters with patient ratings.**

| Cluster | Treatment Goals | % Very Important (n = 43) | Importance Rating[a] | |
|---|---|---|---|---|
| | | | M | SD |
| 1 | **To have trust in the doctor, medical team, and hospital** | | **3.76** | **0.43** |
| | To have a good medical team and facility. | 90.70 | 3.91 | 0.29 |
| | To have trust in my doctor. | 88.37 | 3.88 | 0.32 |
| | Good communication with my doctor and heart team. | 81.40 | 3.81 | 0.39 |
| | To know what my doctor recommends. | 53.49 | 3.44 | 0.70 |
| 2 | **To have good information about my treatment options** | | **3.70** | **0.50** |
| | To receive accurate information about the treatment and possible complications. | 83.72 | 3.84 | 0.37 |
| | To be aware of available options. | 60.47 | 3.56 | 0.63 |
| 3 | **To lead a long life** | | **3.51** | **0.91** |
| | To lead a long life. | 72.09 | 3.51 | 0.91 |
| 4 | **To reduce fear of heart failure and future risks** | | **3.47** | **0.95** |
| | To reduce fear of heart failure. | 67.44 | 3.51 | 0.86 |
| | To find a long-lasting solution to avoid repeat treatment. | 67.44 | 3.47 | 0.96 |
| | To choose a treatment that has fewer long-term risks. | 67.44 | 3.44 | 1.03 |
| 5 | **To improve my health, breathing, and quality of life** | | **3.45** | **0.88** |
| | To improve my health. | 67.44 | 3.60 | 0.62 |
| | To improve my quality of life. | 62.79 | 3.51 | 0.80 |
| | To have more energy, strength, and stamina. | 53.49 | 3.42 | 0.79 |
| | To breathe without difficulty. | 65.12 | 3.28 | 1.30 |
| 6 | **To feel comfortable about my treatment decision, medications, and my future plans** | | **3.40** | **0.94** |
| | Feeling confident that I made the right decision. | 65.12 | 3.49 | 0.91 |
| | To feel comfortable with the medications prescribed. | 53.49 | 3.37 | 0.93 |
| | To be able to make realistic plans for the rest of my life. | 55.81 | 3.33 | 0.97 |
| 7 | **To be independent or active** | | **3.29** | **0.95** |
| | To be independent. | 72.09 | 3.67 | 0.61 |
| | To be able to do my normal activities. | 76.74 | 3.65 | 0.75 |
| | To be physically active. | 65.12 | 3.51 | 0.83 |
| | To lead an active lifestyle. | 62.79 | 3.49 | 0.83 |
| | To be able to travel. | 46.51 | 3.09 | 1.17 |
| | To be able to work. | 27.91 | 2.33 | 1.51 |
| 8 | **To spend time with family and give them peace of mind** | | **3.06** | **1.20** |
| | To spend time with family and friends. | 44.19 | 3.16 | 1.04 |
| | To give my family peace of mind. | 53.49 | 2.95 | 1.36 |
| 9 | **To have a less invasive procedure, shorter recovery, and to know what to expect and ensure support services** | | **2.96** | **1.15** |
| | To avoid open heart surgery by choosing a minimally invasive procedure. | 58.14 | 3.14 | 1.30 |
| | To know what to expect for recovery and ensure that support services are in place. | 53.49 | 3.37 | 0.85 |
| | To minimize the length of recovery. | 46.51 | 3.26 | 0.98 |
| | To avoid general anesthesia. | 20.93 | 2.07 | 1.47 |
| 10 | **Other concerns: Covid, costs, ethics** | | **2.03** | **1.57** |
| | To avoid getting Coronavirus or other infections by having the procedure. | 44.19 | 2.42 | 1.71 |
| | To be aware of the cost. | 20.93 | 2.02 | 1.47 |
| | To choose an ethical treatment (avoid animal tissue valve). | 16.28 | 1.65 | 1.54 |

[a] Importance ratings: 4 = "*Very Important*"; 3 = "*Important*"; 2 = "*Neutral*"; 1 = "*Slightly Important*"; 0 = "*Not Important or Does not Apply*"

[b]TAVR = Transcatheter Aortic Valve Replacement.

[c]SAVR = Surgical Aortic Valve Replacement.

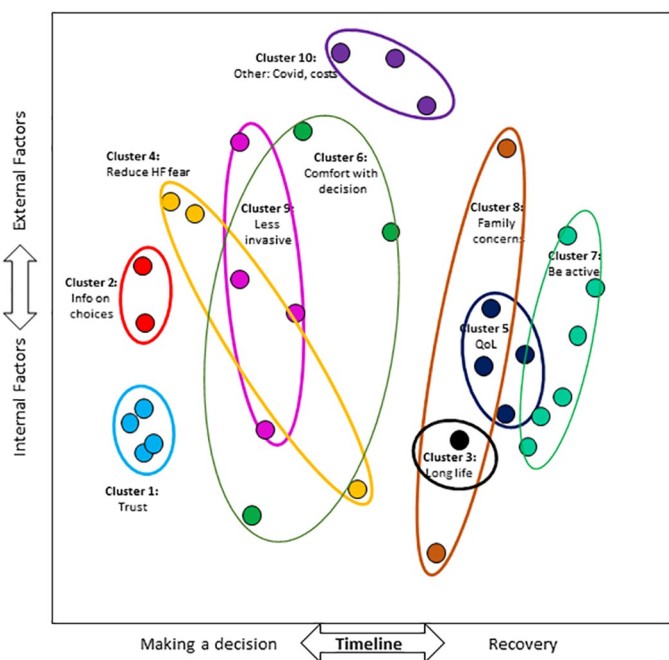

**Fig 2. Cognitive map of patient treatment goals.** Each cluster identifies how specific treatment goals identified using the nominal group technique were grouped and labeled by a group of 38 patients during card sorting activities. Multidimensional scaling defined the spatial orientation of each specific goal and hierarchical cluster analysis guided the identification of the clusters. HF denotes heart failure. Info denotes information. QoL denotes quality of life.

## Quality of life

Quality of life is an important global measure used to compare treatments in controlled studies and benefit-risks analyses. Differences in how quality of life-related concepts are defined matter. Our findings suggest that the global construct *quality of life* conveys different meanings when used by patients than when assessed in clinical trials. While many of the patient-reported items identified might be loosely interpreted as being encompassed by *quality of life*, patients reported *Improving my quality of life* as a distinct goal that clustered with *To breathe without difficulty*, *To improve my health*, and *To have more energy, strength and stamina*. Patients organized concepts such as *Being independent or active* and *Spending time with family* distinct from quality of life. Disaggregating quality of life-related concepts would place more weight on quality of life and could potentially shift the balance of benefits and risks in clinical trials and influence decisions about treatment.

**The physician-patient relationship.** The primacy of *trust in the doctor and medical team* reported by our participants builds upon previous studies showing the significance of patient trust in decision-making [7]. In one study, patients lacking trust in their physician felt ambivalent about TAVR, with some seeking second opinions to find a physician they could trust [25]. Those who trusted their physician accepted the risks of TAVR and felt confident about their decision [25]. Although our study did not assess factors that build trust, trust in one's doctor was a distinct item that clustered closely with *good communication*, *physician experience*, *hospital reputation*, and knowing their *doctor's recommendation*. Prior studies found that trust is not a given but has to be established through relationships and behaviors [26, 27]. Specifically, patients' trust can be earned by spending time sharing information [27], having respectful relationships, and giving patients the opportunity to think through their decision [26–28].

**Table 3. Patient treatment features, prioritized clusters with patient ratings.**

| Cluster | Treatment features | % Very important (n = 41) | Importance rating[a] | |
|---|---|---|---|---|
| | | | M | SD |
| 1 | **Qualities of the doctor and hospital (experience, trust, reputation)** | | **3.82** | **0.45** |
| | How experienced is the physician who will be doing the procedure? | 87.80 | 3.85 | 0.42 |
| | How much faith or trust do I have in the physician doing the procedure? | 85.37 | 3.83 | 0.44 |
| | What is the reputation and experience of the hospital doing the procedure on people like me? | 80.49 | 3.78 | 0.47 |
| 2 | **Comparing my options (overall success, doctor's recommendation)** | | **3.60** | **0.59** |
| | What are the options when replacing or repairing my valve? | 92.68 | 3.93 | 0.26 |
| | Is one procedure more successful in the long term? | 58.54 | 3.54 | 0.60 |
| | Which valve treatment does my doctor recommend and why? | 65.85 | 3.54 | 0.81 |
| | Why would someone choose one procedure over another (SAVR[b] or TAVR[c])? | 53.66 | 3.41 | 0.71 |
| 3 | **Long-term outlook (lifespan, heart failure, repeat procedure)** | | **3.46** | **0.73** |
| | Will I have a better chance of living longer after replacing my valve? | 78.05 | 3.73 | 0.55 |
| | If a second procedure is needed, what will it be and when will it happen? | 56.10 | 3.44 | 0.74 |
| | Will my risk of heart failure increase as I age? | 43.90 | 3.22 | 0.91 |
| 4 | **If and when to have the procedure (risks and urgency)** | | **3.45** | **0.92** |
| | What are the risks of not doing a valve replacement? | 78.05 | 3.73 | 0.55 |
| | What are all the risks involved during and after the procedure (including death)? | 85.37 | 3.73 | 0.78 |
| | How soon should the procedure be done and why? | 56.10 | 3.37 | 0.97 |
| | What are the risks of postponing valve replacement during the COVID-19 pandemic? | 48.78 | 2.95 | 1.40 |
| 5 | **Impact on my daily living and quality of life** | | **3.26** | **0.88** |
| | How will the procedure affect my quality of life? | 68.29 | 3.63 | 0.62 |
| | Will I be able to return to my normal activities and routines, and when? | 48.78 | 3.29 | 0.90 |
| | Will I need to change my daily living habits, such as diet and exercise? | 36.59 | 3.05 | 1.00 |
| | How will I feel after the procedure, physically and emotionally? | 36.59 | 3.05 | 1.00 |
| 6 | **Impact on my overall health and medications** | | **3.20** | **1.04** |
| | How will a valve replacement affect my overall state of health, such as clotting, immunity, and COVID-19? | 75.61 | 3.63 | 0.73 |
| | How will being on a blood thinner affect my life? | 48.78 | 3.20 | 1.08 |
| | Will the procedure help me breathe better? | 51.22 | 3.05 | 1.26 |
| | Will I need to change my medications after the procedure? | 39.02 | 2.93 | 1.10 |
| 7 | **Details about the procedures, processes, and complications** | | **3.20** | **1.01** |
| | What other heart problems might occur as a result of replacing my aortic valve, such as atrial fibrillation, stroke, or need for a pacemaker? | 75.61 | 3.68 | 0.65 |
| | What precautions are taken to reduce the risk of stroke during the procedure? | 75.61 | 3.61 | 0.86 |
| | What is the back-up plan if there are complications during the procedure? | 73.17 | 3.56 | 0.90 |
| | How invasive is the procedure? | 51.22 | 3.15 | 1.15 |
| | What is the expected process from admission to discharge? | 39.02 | 2.98 | 1.04 |
| | Will I feel pain or discomfort during or after the procedure? | 36.59 | 2.95 | 1.16 |
| | How is anesthesia given and will I be awake? | 34.15 | 2.88 | 1.12 |
| | What are the details of the procedure (function, placement, and how it looks)? | 34.15 | 2.76 | 1.18 |
| 8 | **Recovery and post-surgery care (hospital stay, rehab, home care)** | | **2.93** | **1.01** |
| | What will recovery be like? | 39.02 | 3.15 | 0.91 |
| | Will I need cardiac rehab after the procedure? | 29.27 | 2.93 | 0.93 |
| | What type of care will I need at home? | 29.27 | 2.88 | 1.03 |
| | What's the length of stay in the hospital? | 29.27 | 2.76 | 1.16 |

(*Continued*)

**Table 3.** (Continued)

| Cluster | Treatment features | % Very important (n = 41) | Importance rating[a] | |
|---|---|---|---|---|
| | | | M | SD |
| 9 | **Why I have aortic stenosis and how that factors into the treatment decision** | | **2.80** | **1.10** |
| | How does my health history factor into the treatment decision? | 53.66 | 3.41 | 0.74 |
| | What caused my aortic valve stenosis? | 24.39 | 2.63 | 1.16 |
| | Is my aortic stenosis hereditary? | 21.95 | 2.37 | 1.39 |
| 10 | **Details about the new valve (materials, durability, benefit to people with chest radiation)** | | **2.80** | **1.21** |
| | How long will the new valve last? | 43.90 | 3.56 | 0.63 |
| | What are the valves made of (animal tissue or mechanical)? | 43.90 | 2.85 | 1.33 |
| | How many people have had TAVRs to date? | 24.39 | 2.61 | 1.18 |
| | How do the benefits and risks of the procedure compare for someone who had chest radiation treatments? | 31.71 | 2.17 | 1.70 |
| 11 | **Other considerations (time to reflect on my decision, anxiety about aortic stenosis, cost)** | | **2.49** | **1.24** |
| | Will I have time to reflect on my treatment decision and discuss it with my family? | 39.02 | 2.95 | 1.14 |
| | How do I ease my fear or anxiety about aortic stenosis and choosing treatment? | 36.59 | 2.83 | 1.22 |
| | What's the cost and how much will I have to pay? | 29.27 | 2.49 | 1.34 |
| | Can I hear firsthand experiences from other patients? | 14.63 | 2.24 | 1.26 |
| | How long before I can resume air travel? | 9.76 | 1.93 | 1.23 |

[a]Importance ratings: 4 = "*Very Important*"; 3 = "*Important*"; 2 = "*Neutral*"; 1 = "*Slightly Important*"; 0 = "*Not Important or Does not Apply*"

[b]TAVR = Transcatheter Aortic Valve Replacement.

[c]SAVR = Surgical Aortic Valve Replacement.

Notably, these trust-building strategies were mirrored by our participants (e.g., the importance of *accurate information*, *good communication*, and *time to reflect*). As these items are also key elements of SDM [29], building trust may be an additional benefit of engaging in SDM.

### Shared decision-making processes

Our findings confirm the difficulty of decision-making for people with sAS, with over a third strongly endorsing *How do I ease my fear or anxiety about AS and choosing treatment*? Stress can interfere with decision-making, impair perceptions of the benefits (versus risks) and shift people away from goal-directed decision-making [30]. This can lead people to defer decision-making to others or avoid decision-making by defaulting to the status quo (e.g., no AVR) [31]. Additionally, people facing difficult trade-offs between dominant options (e.g., SAVR and TAVR) may choose an inferior default option (e.g., no AVR) simply to avoid the stress of making uncomfortable trade-offs [31, 32]. The finding that 78% of our participants wanted to understand the risks of not doing AVR suggests that many considered this default option. Half of our participants thought it important to know how soon the procedure needed to be done and whether it could be postponed, suggesting decision avoidance. Decision aids have been shown to lower decisional conflict and improve communication [33]; better communication can lower the anxiety of decision-making and help people engage in decision-making [34].

Patient treatment preferences have been implicated as the dominant explanation for the underutilization of AVR. In one study [35], physicians reported patients' preference as the reason for selecting medical care in 31% of cases, followed by medical futility (19.7%) and inoperability/anatomic infeasibility (11.3%). However, patients pursuing medical care reported having insufficient education about their options and feeling uncertain about their decision

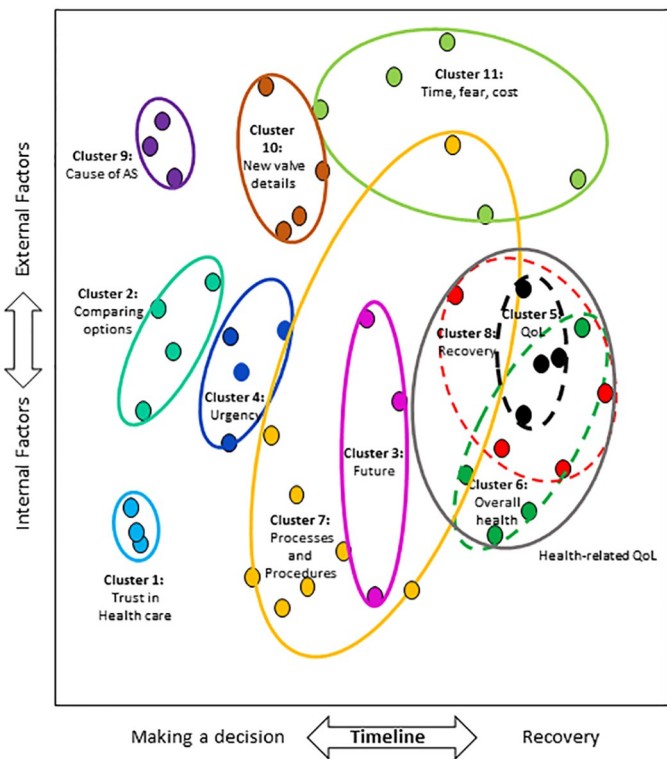

**Fig 3. Cognitive map of patient treatment features.** Each cluster identifies how the specific treatment features identified using the nominal group technique were grouped and labeled by 36 patients during card sorting activities. Multidimensional scaling defined the spatial orientation of each specific feature, hierarchical cluster analysis guided the identification of clusters. QoL denotes quality of life.

[35]. Attributing underutilization of AVR to patient preferences is complicated by our finding that patients' treatment preferences are influenced by their provider's recommendation, with over half (65.9%) reporting that their doctor's recommendation was "very important" to them. Disentangling physician preferences from patient preferences is challenging when patient preferences are shaped by their physician's recommendations.

To make a recommendation about treatment, HCPs must weigh the benefits and risks of treatment options based on the scientific evidence, which requires making value judgments about which benefits and risks to consider and their relative importance. These judgments have traditionally been made by physicians based on endpoints defined in clinical trials [36], which historically have not reflected the perspectives and values of patients. If patient values are not assessed, these judgments may be based on physicians' assumptions about what they perceive matters to their patients. Yet physicians are often inaccurate when making substituted judgments for patients [37, 38]. Surrogate decision-makers may confuse their own or others' interests with the patient's interest [38]. Assumptions about what matters to patients may also influence physician judgements about medical futility. Studies in other areas found decisions about withdrawal of care to be subject to bias by the prognosticating physicians, who offered certain treatment options at their own discretion [39]. Further, withdrawal of care (e.g., no AVR) can be a self-fulfilling prophecy that can reinforce implicit biases about futility [40]. Just as patients' treatment preferences are influenced by their impressions of their medical team, physicians' treatment recommendations may be influenced by their impressions of their patients and their assumptions about what matters to them.

**Shared decision-making processes in minority populations.**   Racial minorities are underrepresented among patients undergoing AVR [41, 42]. Blacks appear less likely to receive a physician recommendation for AVR (compared to non-Blacks) [43] despite similar survival post-AVR [43], suggesting that implicit bias may influence treatment decisions [44]. When AVR is recommended, Blacks are more likely to refuse it [43, 45], which may reflect lack of trust based on previous negative experiences or the absence of minority representation in medical settings. Accepting a patient's preference for treatment at face value without understanding their goals, values, and reasoning, or attempting to correct any misperceptions, can perpetuate inequities in care. A more standardized approach to SDM that assesses patient priorities and educates patients about their treatment alternatives should lead to more appropriate and equitable care [46, 47].

Despite the increasing emphasis on SDM in clinical guidelines [2], there are no accepted practices for identifying patients' goals and values in clinical, research, or regulatory settings [48]. People facing complex, unfamiliar, or stressful situations often do not have clear ideas about what matters to them and have difficulty applying their values to decisions [49]. Several approaches have been used to help people determine what matters to them when facing a decision, including values clarification exercises [50], stated preference approaches (e.g., discrete choice experiments, conjoint analyses) and utility theory (e.g., risk-benefit analyses). All these approaches involve selecting a specific set of treatment attributes and outcomes and assigning weights to them. While much attention has been paid to the methods for assigning weights, scant attention has been paid to how the attributes were selected, which is often entrusted to the investigators with little transparency or patient input [51]. The attributes that are chosen can drive results; choosing the wrong set can lead to spurious findings. The methods used in this study provide a transparent patient-centered approach to eliciting patient-reported goals and values.

The patient-reported goals and features identified in our study can inform the attributes that are included in patient preference information studies to improve their validity [48]. Those studies can inform the clinical endpoints that are included in controlled studies so that robust scientific evidence can be garnered on patient-identified endpoints [48]. The evidence generated can make benefit-risk analyses more patient-centered by including items that reflect how patients consider benefit-risk trade-offs. Many of our patient-reported items do not directly map to the evidence from controlled studies because the data measured in these studies have only included items chosen by investigators. These items can also be used to guide the topics included in decision aids. We are using these items to develop a SDM tool to help patients with sAS clarify their goals, values, and preferences for treatment and share this information with their HCP.

Helping patients clarify and communicate their goals and values with their HCP should help them arrive at an informed treatment preference that is consistent with their own priorities and help HCPs adhere to clinical guidelines recommending SDM [2]. Making HCPs aware of what matters to patients can help focus discussions on what matters to them, making it easier for patients to process information and participate in decision-making [11, 52]. Tailoring treatment discussions to what matters to patients is essential for SDM [52] and can help HCPs choose a treatment that aligns with their patient's values.

Our study has several limitations. Our sample was small albeit appropriate for the methods employed. Over-representation of educated white people limits generalizability, though 20% of participants had low health literacy. The high ratings assigned to many items limit our ability to explore differences in ratings but confirms the relevance of the items reported. The methods employed were designed to identify key patient goals and treatment features; other methods applied to a larger sample are more appropriate for prioritization. We combined

findings from online and in-person activities, recognizing that the method of collecting responses may influence those responses. Asynchronous online activities support more inclusive sampling and thoughtful responses by removing time constraints. The labels assigned to the MDS dimensions are speculative. These methods minimize but do not eliminate investigator bias.

## Conclusions

Decisions about treating sAS are influenced by a broad range of treatment goals and features that vary substantially across individuals and that diverge from outcomes included in clinical trials. Our findings challenge the validity of using investigator-identified features to understand or guide patient decisions. Helping patients with sAS clarify their goals and preferences and share them with their HCPs may improve the patient-centeredness of care.

## Supporting information

**S1 Appendix. Supporting methods, recruitment, study procedures, statistical analyses, and S1-S4 Figs.**
(DOCX)

## Acknowledgments

We gratefully acknowledge the contributions of Dale Boisvert and Gary and Robin Townes.

## Author Contributions

**Conceptualization:** Nananda F. Col, Long Ngo, Megan Coylewright.

**Data curation:** Nananda F. Col, Diana Otero, Long Ngo, Kimberly Goodloe, Susan Strong, Elvin Kaplan, Melissa Beaudry, Megan Coylewright.

**Formal analysis:** Nananda F. Col, Long Ngo.

**Funding acquisition:** Nananda F. Col, Megan Coylewright.

**Investigation:** Nananda F. Col, Megan Coylewright.

**Methodology:** Nananda F. Col, Long Ngo.

**Project administration:** Nananda F. Col, Diana Otero, Brian R. Lindman, Aaron Horne, Melissa M. Levack, Kimberly Goodloe, Megan Coylewright.

**Resources:** Nananda F. Col, Diana Otero, Brian R. Lindman, Aaron Horne, Melissa M. Levack, Susan Strong, Elvin Kaplan, Melissa Beaudry, Megan Coylewright.

**Software:** Nananda F. Col.

**Supervision:** Nananda F. Col, Long Ngo, Megan Coylewright.

**Validation:** Nananda F. Col, Kimberly Goodloe, Elvin Kaplan.

**Visualization:** Nananda F. Col.

**Writing – original draft:** Nananda F. Col, Diana Otero, Brian R. Lindman, Aaron Horne, Melissa M. Levack, Long Ngo, Kimberly Goodloe, Susan Strong, Elvin Kaplan, Melissa Beaudry, Megan Coylewright.

**Writing – review & editing:** Nananda F. Col, Diana Otero, Brian R. Lindman, Aaron Horne, Melissa M. Levack, Long Ngo, Kimberly Goodloe, Susan Strong, Elvin Kaplan, Melissa Beaudry, Megan Coylewright.

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
