## [Decision Letter · Decision Letter 0]

31 Mar 2022

PONE-D-21-38629What Matters Most to Patients with Severe Aortic Stenosis when Choosing Treatment? Framing the Conversation for Shared Decision MakingPLOS ONE

Dear Dr. Nananda Col,

Thank you for submitting your manuscript to PLOS ONE. After careful consideration, we feel that it has merit but does not fully meet PLOS ONE’s publication criteria as it currently stands. Therefore, we invite you to submit a revised version of the manuscript that addresses the points raised during the review process.

The reviewers acknowledge the clinical importance of your work. Only minor revision is required.

We look forward to receiving your revised manuscript.

Kind regards,

Cécile Oury

Academic Editor

PLOS ONE

Journal Requirements:

Reviewers' comments:

Reviewer's Responses to Questions

**Comments to the Author**

1. Is the manuscript technically sound, and do the data support the conclusions?

Reviewer #1: Yes

2. Has the statistical analysis been performed appropriately and rigorously? 

Reviewer #1: Yes

3. Have the authors made all data underlying the findings in their manuscript fully available?

Reviewer #1: Yes

4. Is the manuscript presented in an intelligible fashion and written in standard English?

Reviewer #1: Yes

5. Review Comments to the Author

Reviewer #1: In their study entitled “What Matters Most to Patients with Severe Aortic Stenosis when Choosing Treatment? Framing the Conversation for Shared Decision Making,” Col and colleagues aim to “identify, prioritize, and organize patient-reported goals and features of treatment” for severe aortic stenosis. They perform a very rigorous study which attempts to minimize investigator bias and identify treatment goals and features, clustering them and assessing associations in an effort to lay the groundwork for the development of shared decision-making tools and relevant patient-focused outcomes in clinical trials.

In their introduction, the authors of this study clearly and succinctly frame the issues and highlight the lack of understanding of patient-reported goals and features of treatment in the literature. As the authors point out, this is an important study which provides information necessary for the development of tools to guide decision-making and future study design, and as such I think it makes an important contribution to the literature. It is also quite timely as there is recent widespread interest in understanding the reasons for the clear undertreatment of AS and the racial and ethnic disparities in treatment. Patient goals and patient-defined treatment features likely play a significant role.

I am an interventional cardiologist with a busy TAVR practice and found the results of this study interesting and illuminating, and in some cases surprising. The results of this study are immediately useful to clinicians who are having these conversations with patients every day, even as we await the authors’ more rigorous extension of this preliminary research.

The methods are well-described and seem appropriate. This research methodology is not familiar to me and seems very particular to this type of research, so the editors could consider adding a reviewer with expertise in this methodology if they have not already.

The results are clearly presented, and the figures complement the text.

The discussion puts the results in appropriate context and the conclusions are appropriate.

Overall, I think the authors are to be congratulated on a well-done study which is immediately relevant to clinical practice and addresses a wide gap in the literature.

Minor comments:

1. Page 2, line 2: “NGT” is not defined prior to its first use.

2. Page 3, line 4: “WCG” is not defined.

3. Could the authors expand a bit more on how the change to virtual groups (rather than in-person) may have affected their results?

4. Can the authors please specify the repository for their data and insert a statement into the methods section about its availability and location?

6. PLOS authors have the option to publish the peer review history of their article (what does this mean?). If published, this will include your full peer review and any attached files.

Reviewer #1: No

---

## [Author Response · Author response to Decision Letter 0]

12 Apr 2022

Response to Reviewer #1

Minor comments:

Comment 1. Page 2, line 2: “NGT” is not defined prior to its first use.

Response: This oversight has been corrected. NGT stands for Nominal Group technique. (Page 2, line 2)

Comment 2. Page 3, line 4: “WCG” is not defined.

Response: This has been corrected. Western Institutional Review Board (WIRB) Copernicus Group (WCG) Independent Review Board. (Page 3, lines 6-7)

Comment 3. Could the authors expand a bit more on how the change to virtual groups (rather than in-person) may have affected their results?

Response: The change to virtual groups should not have substantially affected our results because we had previously validated the similarity between the in-person and online protocols. Because we use the same video recordings of instructions for each component of the NGT, the 2 protocols closely mirror each other.

We added a phrase to the methods section to establish the similarities between the online and in-person protocols: “An online asynchronous NGT protocol that mirrored the in-person protocol had been previously developed and validated.” (Page 4, lines 3-4). 

We also added 2 sentences in the Discussion section: We combined findings from online and in-person activities, recognizing that the method of collecting responses may influence those responses. Asynchronous online activities support more inclusive sampling and thoughtful responses by removing time constraints.” (Page 18, lines 17-20)

Comment 4. Can the authors please specify the repository for their data and insert a statement into the methods section about its availability and location?

Response: We would like to make the data available through the Dryad depository, if possible. We added a statement as requested: “The data underlying our findings are available through the Dryad depository.” (Page 4, line 6.) I have never used this depository before and do not know how to access it, as it appears to be limited to members. If Dryad is not available, I can post the data through whatever mechanism is recommended.

---

## [Editor Report · Decision Letter 1]

7 Jun 2022

What Matters Most to Patients with Severe Aortic Stenosis when Choosing Treatment? Framing the Conversation for Shared Decision Making

PONE-D-21-38629R1

Dear Dr. Nananda Col,

We’re pleased to inform you that your manuscript has been judged scientifically suitable for publication and will be formally accepted for publication once it meets all outstanding technical requirements.

Kind regards,

Cécile Oury

Academic Editor

PLOS ONE
---

## [Editor Report · Acceptance letter]

3 Aug 2022

PONE-D-21-38629R1 

What Matters Most to Patients with Severe Aortic Stenosis when Choosing Treatment? Framing the Conversation for Shared Decision Making 

Dear Dr. Col:

I'm pleased to inform you that your manuscript has been deemed suitable for publication in PLOS ONE. Congratulations! Your manuscript is now with our production department. 

Kind regards, 

on behalf of

Dr. Cécile Oury 

Academic Editor

PLOS ONE